# Tips for Hepatologist Referral of Patients with Metabolic Dysfunction-Associated Steatotic Liver Disease with Alanine Aminotransferase Levels ≤ 30 U/L

**DOI:** 10.3390/diagnostics15131591

**Published:** 2025-06-23

**Authors:** Miwa Kawanaka, Hideki Fujii, Michihiro Iwaki, Hideki Hayashi, Hidenori Toyoda, Satoshi Oeda, Hideyuki Hyogo, Asahiro Morishita, Kensuke Munekage, Kazuhito Kawata, Tsubasa Tsutsumi, Koji Sawada, Tatsuji Maeshiro, Hiroshi Tobita, Yuichi Yoshida, Masafumi Naito, Asuka Araki, Shingo Arakaki, Takumi Kawaguchi, Hidenao Noritake, Masafumi Ono, Tsutomu Masaki, Satoshi Yasuda, Eiichi Tomita, Masato Yoneda, Akihiro Tokushige, Yoshihiro Kamada, Hirokazu Takahashi, Shinichiro Ueda, Shinichi Aishima, Ken Nishino, Katsunori Ishii, Takashi Fushimi, Hirofumi Kawamoto, Yoshio Sumida, Takeshi Okanoue, Atsushi Nakajima

**Affiliations:** 1Department of Gastroenterology and Hepatology, Okayama University Graduate School of Medicine, Okayama 700-8558, Japan; 2Department of General Internal Medicine 2, Kawasaki Medical Center, Kawasaki Medical School, Okayama 700-8505, Japan; k-nishino@med.kawasaki-m.ac.jp (K.N.); katsunori.ishii@med.kawasaki-m.ac.jp (K.I.); fushimi_takashi@med.kawasaki-m.ac.jp (T.F.); hirofumi.kawamoto@gmail.com (H.K.); 3Department of Hepatology, Graduate School of Medicine, Osaka Metropolitan University, Osaka 545-8586, Japan; fujiirola@yahoo.co.jp; 4Department of Gastroenterology and Hepatology, Yokohama City University Graduate School of Medicine, Yokohama 232-0024, Japan; michihirokeidai@yahoo.co.jp (M.I.); dryoneda@yahoo.co.jp (M.Y.); nakajima-tky@umin.ac.jp (A.N.); 5Department of Gastroenterology and Hepatology, Gifu Municipal Hospital, Gifu 500-8513, Japan; hidekihaya884@gmail.com (H.H.); etomita_jp@yahoo.co.jp (E.T.); 6Department of Gastroenterology and Hepatology, Ogaki Municipal Hospital, Ogaki 503-8502, Japan; hmtoyoda@spice.ocn.ne.jp (H.T.); satoshi.yasuda.1982@gmail.com (S.Y.); 7Liver Center, Saga University Hospital, Saga 849-8501, Japan; ooedasa@cc.saga-u.ac.jp (S.O.); takahas2@cc.saga-u.ac.jp (H.T.); 8Hyogo Life Care Clinic Hiroshima, Hiroshima 732-0823, Japan; hidehyogo@ae.auone-net.jp; 9Department of Gastroenterology and Neurology, Faculty of Medicine, Kagawa University, Takamatsu 761-0793, Japan; morishita.asahiro@kagawa-u.ac.jp (A.M.); tmasaki@med.kagawa-u.ac.jp (T.M.); 10Department of Gastroenterology, Kochi Prefectural Hata Kenmin Hospital, Kochi 788-0785, Japan; jm-k.munekage@kochi-u.ac.jp; 11Hepatology Division, Department of Internal Medicine II, Hamamatsu University School of Medicine, Shizuoka 431-3192, Japan; kawata@hama-med.ac.jp (K.K.); noritake@hama-med.ac.jp (H.N.); 12Division of Gastroenterology, Department of Medicine, Kurume University School of Medicine, Kurume 830-0011, Japan; tsutsumi_tsubasa@med.kurume-u.ac.jp (T.T.); takumi@med.kurume-u.ac.jp (T.K.); 13Division of Gastroenterology, Department of Internal Medicine, Asahikawa Medical University, Asahikawa 078-8510, Japan; k-sawada@asahikawa-med.ac.jp; 14First Department of Internal Medicine, University of the Ryukyus Hospital, Okinawa 903-0215, Japan; tajjimaeshiro@gmail.com (T.M.); h052010@med.u-ryukyu.ac.jp (S.A.); 15Department of Hepatology, Shimane University Hospital, Izumo 693-8501, Japan; ht1020@med.shimane-u.ac.jp; 16Department of Gastroenterology and Hepatology, Suita Municipal Hospital, Osaka 564-8567, Japan; yu1yoshida@gmail.com (Y.Y.); naito0757@mhp.suita.osaka.jp (M.N.); 17Pathology Division, Shimane University Hospital, Izumo 693-8501, Japan; asuka@med.shimane-u.ac.jp; 18Division of Innovative Medicine for Hepatobiliary & Pancreatology, Faculty of Medicine, Kagawa University, Takamatsu 760-0793, Japan; ono.masafumi@kagawa-u.ac.jp; 19Department of Clinical Pharmacology, Therapeutics School of Medicine, University of the Ryukyus, Okinawa 903-0213, Japan; akihiro@med.u-ryukyu.ac.jp (A.T.); blessyou@med.u-ryukyu.ac.jp (S.U.); 20Department of Advanced Metabolic Hepatology, Osaka University Graduate School of Medicine, Osaka 565-0871, Japan; 21Department of Scientific Pathology, Graduate School of Medical Sciences, Kyushu University, Fukuoka 812-8582, Japan; aishima.shinichi.476@m.kyushu-u.ac.jp; 22Graduate School of Healthcare Management, International University of Healthcare and Welfare, Tokyo 107-8402, Japan; sumida19701106@yahoo.co.jp; 23Department of Gastroenterology and Hepatology, Saiseikai Suita Hospital, Osaka 564-0013, Japan; okanoue@suita.saiseikai.or.jp

**Keywords:** alanine transaminase, FIB-4 index, MASLD, Nara Declaration 2023, diabetes mellitus type-2

## Abstract

**Background/Objectives:** The possibility of progressive liver fibrosis remains even when alanine aminotransferase (ALT) levels are <30 IU/L. Therefore, we aimed to investigate factors that can predict fibrosis progression in patients with metabolic dysfunction-associated steatotic liver disease (MASLD) with ALT levels ≤ 30 U/L. **Methods:** This multicenter retrospective cohort study was conducted using data collected between December 1994 and December 2021. Among the 1381 patients with MASLD (CLIONE study) who underwent liver biopsy, we performed decision-tree analysis on factors for stage ≥ 3 in 115 with ALT levels ≤ 30 U/L. Of the 818 patients with MASLD (Kawasaki cohort) who underwent liver biopsy, we included 174 with ALT levels ≤ 30 U/L for validation. **Results:** In the decision-tree analysis of patients with stage ≥ 3 with ALT levels ≤ 30 U/L, 57% of patients with a fibrosis-4 (FIB-4) index ≥ 2.67 and 70% with both FIB-4 index ≥ 2.67 and type-2 diabetes mellitus (DM) were detected. However, no cases of stage ≥ 3 were observed among patients without type-2 DM with ALT ≤ 30 U/L and a FIB-4 index < 2.67. After verifying the decision-tree analysis, the model construction and validation datasets showed a close correlation. **Conclusions:** Among patients with MASLD with ALT levels ≤ 30 U/L, those with an FIB-4 index ≥ 2.67, particularly with comorbid type-2 DM, should consider consultation with a hepatologist.

## 1. Introduction

Metabolic dysfunction-associated steatotic liver disease (MASLD) is the most common chronic liver disease (CLD), and its prevalence has recently increased [1,2]. Consequently, early detection of patients with MASLD with poor prognosis is important, and collaboration between primary care physicians and specialists is necessary.

The Japan Society of Hepatology defines alanine aminotransferase (ALT) levels > 30 U/L as a criterion for referral to a gastroenterologist or hepatologist [3]. ALT is an enzyme abundantly present in the cytoplasm of hepatocytes, and serum ALT levels are an important indicator reflecting liver inflammation and damage in patients with CLD. The usefulness of “ALT value > 30 U/L” in patients with MASLD has also been reported and verified [4]. However, patients with MASLD with progressive fibrosis may have ALT levels ≤ 30 U/L. Therefore, we aimed to investigate factors that can predict fibrosis progression in patients with MASLD with ALT levels ≤ 30 U/L.

## 2. Methods

### 2.1. Patients

The CLIONE study included 115 patients with MASLD with ALT levels ≤ 30 U/L of the 1381 with MASLD (age, 63 [range, 28–78] years; male, 29%; liver fibrosis stage 0/1/2/3/4: 40/39/17/13/6, respectively) who underwent liver biopsy at multiple centers between December 1994 and December 2020.

In contrast, the validation study included 174 patients with MASLD with ALT levels ≤ 30 U/L of the 818 with MASLD (age, 61 [range, 18–81] years; male, 39%; liver fibrosis stage 0/1/2/3/4: 38/59/34/34/9, respectively) (Kawasaki cohort) who underwent liver biopsy at Kawasaki Medical School General Medical Center between April 1996 and March 2024 (Table 1).

### 2.2. Procedure

Background factors and blood tests (age, sex, type-2 diabetes mellitus [type-2 DM], hypertension, dyslipidemia, ALT, aspartate aminotransferase [AST], γ-glutamyl transpeptidase, total cholesterol, platelet count, albumin, glycated hemoglobin [HbA1c], fasting blood sugar [FBS], and fibrosis-4 [FIB-4] index) were compared among the 289 patients with MASLD with ALT levels ≤ 30 U/L in both cohorts combined. In addition, decision-tree analysis was performed for stage ≥ 3 factors among 115 patients with MASLD with ALT levels ≤ 30 U/L in the CLIONE study, and 174 patients with MASLD with ALT levels ≤ 30 U/L in the Kawasaki cohort were used to validate the decision-tree analysis.

The MASLD criteria [5] include the presence of steatotic liver disease and the absence of other liver diseases, such as hepatitis B or C virus infection, autoimmune liver disease, drug-induced liver injury, and metabolic liver disease. Individuals with a history of low alcohol consumption (male, <30 g/day; female, <20 g/day) and those with at least one of the following five risk factors for heart disease were considered: (1) body mass index ≥ 23 kg/m^2^ or waist circumference of ≥94 cm for males and ≥80 cm for females; (2) elevated FBS level (≥100 mg/dL), 2-h postprandial blood sugar levels ≥ 140 mg/dL, HbA1c level ≥ 5.7%, and type-2 DM/use of antidiabetic drugs; (3) high blood pressure (≥135/85 mmHg) or use of antihypertensive drugs; (4) elevated triglyceride levels (≥150 mg/dL) or use of lipid-improving drugs; and (5) low levels of high-density lipoprotein cholesterol (≤40 and ≤50 mg/dL for males and females, respectively). The FIB-4 index was calculated as AST (IU/L) × age (years)/platelet count (10^9^/L) × ALT (IU/L) [6].

### 2.3. Liver Biopsy and Histological Analysis

All liver biopsies were performed using a 16- or 17-gauge biopsy needle under ultrasound guidance or a 14-gauge needle under laparoscopic guidance. After the specimens were fixed in 10% formalin and sectioned, they were stained with Azan and hematoxylin–eosin stains. MASLD stage of liver tissue, lobular inflammation, steatosis, and hepatocellular ballooning grades were determined according to the Brunt and Kleiner classification [7,8]. Histological parameters included fibrosis, inflammation, steatosis, hepatocyte ballooning, and the non-alcoholic fatty liver disease activity score (NAS). The NAS scoring system includes lobular inflammation (grades 0–3), steatosis (grades 0–3), and hepatocellular ballooning (grades 0–2) [7,8]. Furthermore, the liver fibrosis stage was assessed according to the Brunt criteria [7]. Two experienced liver pathologists blinded to patient details conducted the histological examinations.

### 2.4. Statistical Analyses

The JMP software version 16.2 (SAS Institute Inc., Cary, NC, USA) was used for the decision-tree analysis. Specifically, the software searched the analytical database for the factor that most effectively predicted stage ≥ 3. The patients were categorized into two groups (stages 0–2/3–4) according to that predictor. Each group was repeatedly assessed and classified according to this two-choice branching method. For each group, median values were calculated using continuous variables, and comparisons between the two groups were performed using *t*-tests. Finally, a stage ≥ 3 prediction model was created. The suitability and reproducibility of the model were validated by comparing the stage ≥ 3 development rates between the model–derivation and validation groups.

The decision-tree analysis included clinical parameters that showed significant differences between stages 0–2 and 3–4. Cutoff values were determined using the Gini coefficient. We used the observed values for quantitative variables, and branching was based on the values deemed optimal. For univariate analyses, Student’s *t*-test and Fisher’s exact test were used for continuous and categorical variables, respectively. Logistic regression was used for multivariate analysis, and statistical significance was set at *p* < 0.05.

The study protocol complied with the guidelines outlined in the 1975 Helsinki Declaration (as revised in Fortaleza, Brazil, in October 2013). This study was approved by the Institutional Review Board of Saga University and Kawasaki Medical School (approval nos. 2020-04-R-02 and 3864).

## 3. Results

### 3.1. Comparison of Patients with MASLD and Fibrosis Stages 3–4 vs. 0–2 with ALT Levels ≤ 30 U/L

Among the 1381 patients with MASLD, 115 had ALT levels ≤ 30 U/L, and stage ≥ 3 was observed in 19 (17%). In the Kawasaki cohort, which was considered a validation study group, 174 of the 818 patients with MASLD had ALT levels ≤ 30 U/L, and stage ≥ 3 was observed in 43 (24%). Univariate analysis revealed that 289 patients with MASLD with ALT levels ≤ 30 U/L in both cohorts were older than those with stages 0–2, with significant differences in age, type-2 DM, hypertension, AST levels, platelet counts, albumin levels, HbA1c levels, and the FIB-4 index. In the multivariate analysis, age ≥ 65 years, the presence of type-2 DM and hypertension were significant factors (Table 2 and Table 3).

### 3.2. Decision-Tree Analysis of Stage ≥ 3 in Patients with MASLD with ALT Levels ≤ 30 U/L

The Gini coefficients for the FIB-4 index cutoff values of 2.67, 2.5, 2.3, and 3.0 were 0.169629, 0.156343, 0.177281, and 0.189256, respectively. Although the FIB-4 index cutoff value of 2.5 had a better Gini coefficient than that of 2.67, we used the cutoff value of 2.67 because it is better known than 2.5. In the CLIONE study, the decision-tree analysis of patients with stage ≥ 3 with ALT levels ≤ 30 U/L showed that the FIB-4 index > 2.67 was the most promising factor, followed by the presence of type-2 DM. The FIB-4 index > 2.67 was found in 57% of the patients, and the FIB-4 index > 2.67 and the presence of type-2 DM were detected in 70%.

Meanwhile, no stage ≥ 3 cases were observed among patients with ALT levels ≤ 30 U/L, a FIB-4 index ≤ 2.67, and no type-2 DM (Figure 1); the same was true for the Kawasaki cohort examined for validation (Figure 2). The validation of the decision-tree analysis showed a close correlation between the model-building and validation datasets (correlation coefficient r^2^ = 0.8997; Figure 3). In addition, the model-building group was more discriminative and stable than the validation group as verified using the correlation between the cumulative number of cases (%) and the cumulative incidence of fibrosis stage ≥ 3 (%) (Figure 4).

## 4. Discussion

Recently, advancement in viral hepatitis treatment has led to a decline in the prevalence of viral liver diseases associated with elevated ALT levels, such as hepatitis B and C [9]. However, an increasing trend has been observed in the number of patients with lifestyle-related diseases [9], such as steatosis and alcohol-related liver disease. Patients with MASLD and mildly elevated ALT levels are frequently overlooked despite their high risk of fibrosis. In patients with ALT levels > 30 U/L, obesity, type-2 DM, dyslipidemia, hypertension, or steatotic liver disease, along with a platelet count ≤ 200,000 or FIB-4 index ≥ 1.3 is used to predict advanced fibrosis [3], and consultation with a gastroenterologist and hepatologist is recommended. Therefore, this requires educating physicians who are not hepatologists regarding the kind of cases to refer to a gastroenterologist and hepatologist.

The prevalence of MASLD has been reported to be higher in patients with elevated ALT levels, particularly in those with ALT levels > 30 U/L in the Japan Society of Hepatology guideline [3]. In our study, we found that in patients with MASLD who underwent liver biopsy, the detection of stage ≥ 2 fibrosis development was possible in 76–94% of cases by lowering the ALT threshold from >50 to >30 U/L. In addition, the prevalence of stage ≥2 fibrosis development was higher in patients with MASLD who underwent liver biopsy than in all patients who underwent liver biopsy. Patients with elevated ALT levels among those with MASLD have a higher risk of liver fibrosis and hepatocarcinogenesis than those without elevated ALT levels, and changes in ALT levels are associated with alterations in liver histology [10].

However, finding predictive factors for fibrosis development, even in patients with ALT levels ≤ 30 IU/L, is important. Gawrieh et al. [11] reported that approximately 10–20% of the patients with ALT levels ≤30 or ≤40 U/L had fibrosis. Liver biopsies of 330 patients with type-2 DM with sustained ALT levels >20 and >30 U/L in females and males, respectively, showed that metabolic dysfunction-associated steatohepatitis (MASH) and cirrhosis are present in 58% and 10% of cases, respectively, indicating that even cases with low ALT levels have fibrosis progression [12].

The American Association for the Study of Liver Diseases practice guidance states that ALT levels are frequently normal in cases of fibrosis progression in MASH and should not be used alone to rule out the presence of MASH with clinically significant fibrosis [13]. Therefore, given the reports of fibrosis development even with ALT levels ≤30 U/L, we predicted cases of liver fibrosis development among patients with MASLD with ALT levels ≤ 30 U/L. In this study, stage ≥ 3 was observed in 63 (22%) of the 289 MASLD cases with ALT levels ≤ 30 U/L.

Here, 54–57% of patients with MASLD with ALT levels ≤30 U/L had a FIB-4 index >2.67 in both CLIONE and Kawasaki studies, and 64–70% of those with MASLD in both studies had a FIB-4 index >2.67 and type-2 DM. The cases of fibrosis development could be detected. However, the cases of fibrosis in patients with ALT levels ≤30 U/L, a FIB-4 index ≤ 2.67, and no type-2 DM were few, indicating that confirming the FIB-4 index and presence of type-2 DM in patients with MASLD with ALT levels ≤ 30 U/L is important.

This study had some limitations. First, we used a hospital-based cohort to examine the association with histopathological findings. In future studies, validation in a health checkup cohort and non-liver-biopsy cases (patients who underwent magnetic resonance elastography and other similar assessments) should be conducted. Second, the database used in this study does not include imaging findings or medication history of the patients. Therefore, the reasons for the ALT levels ≤ 30 U/L during liver biopsy may include mild fibrosis, mild inflammation, and progressive fibrosis. Third, as a retrospective study with a long inclusion period, there may have been variability in clinical decisions and biopsy indications. However, all liver specimens were evaluated by two experienced hepatopathologists using consistent histological criteria, which reduced diagnostic variability and enhanced the reliability of the histological assessments.

In this study, some patients underwent liver biopsy despite having ALT levels ≤ 30 U/L. Although imaging data and medication history were not available in the database, a biopsy was typically considered when there was clinical suspicion of advanced fibrosis or steatohepatitis based on metabolic risk factors, laboratory data, or non-invasive assessments. These factors likely contributed to the clinical decision to proceed with histological evaluation, even in the presence of normal ALT levels.

## 5. Conclusions

The ALT level proposed in the Nara Declaration 2023 can better detect cases of MASLD with fibrosis and predict MASLD with fibrosis with a worse prognosis when combined with platelet count and the FIB-4 index. Among patients with MASLD with ALT levels ≤ 30 U/L, those with a FIB-4 index > 2.67, particularly with comorbid type-2 DM, should be referred to a hepatologist.

## Figures and Tables

**Figure 1 diagnostics-15-01591-f001:**
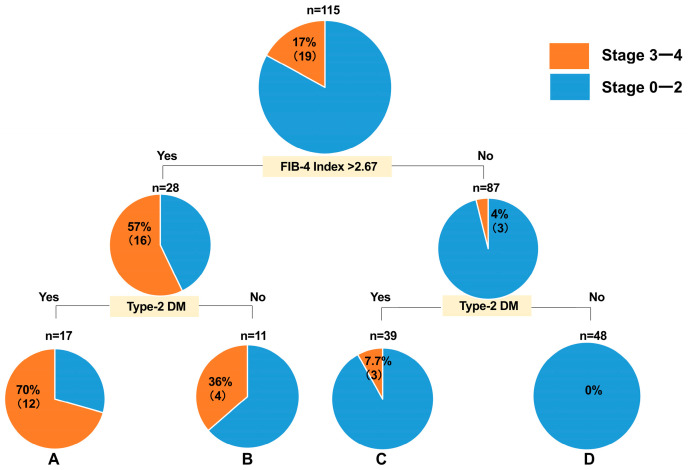
Decision-tree analysis of stage ≥ 3 in patients with ALT levels ≤ 30 U/L in the CLIONE study. ALT: alanine aminotransferase; Type-2 DM: type-2 diabetes mellitus.

**Figure 2 diagnostics-15-01591-f002:**
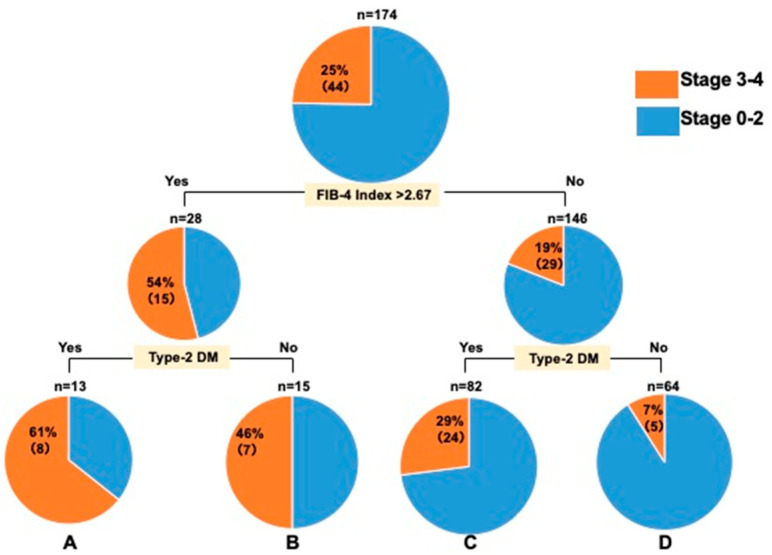
Decision-tree analysis of stage ≥ 3 in patients with ALT levels ≤ 30 U/L in the Kawasaki study. ALT: alanine aminotransferase; Type-2 DM: type-2 diabetes mellitus.

**Figure 3 diagnostics-15-01591-f003:**
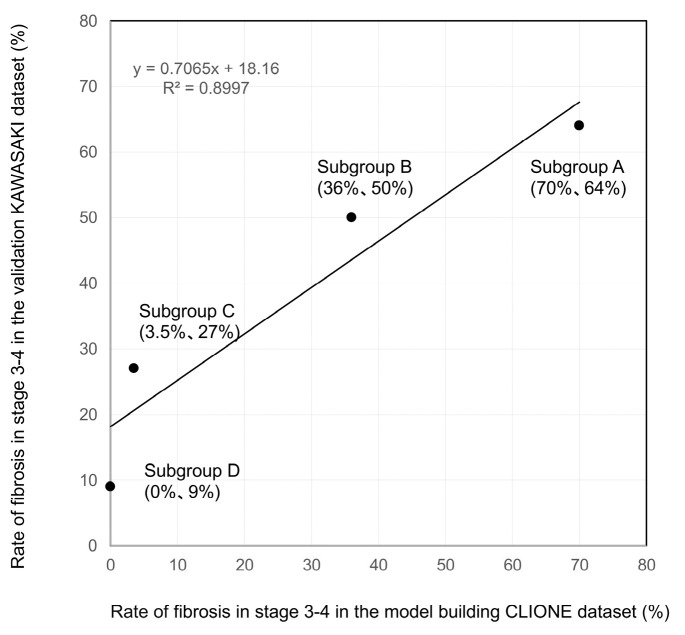
Validation of the decision-tree analysis using an external validation dataset: Sub-group-stratified comparison of the rate of fibrosis stages 3–4. The rate of fibrosis stages 3–4 in each sub-group was plotted. Specifically, the X- and Y-axes represent the model-building and validation datasets, respectively. The model-building and validation datasets (correlation coefficient r^2^ = 0.8997) are closely correlated.

**Figure 4 diagnostics-15-01591-f004:**
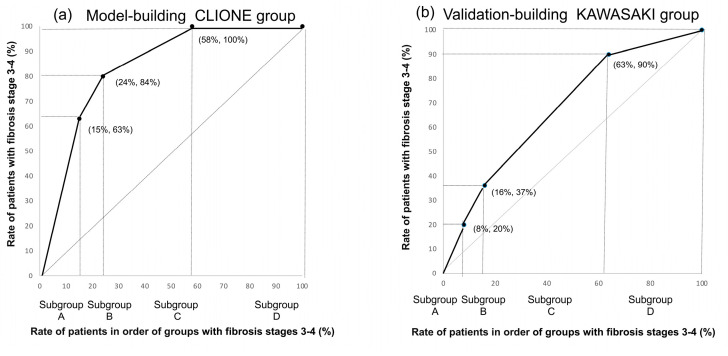
Validation of the efficiency and stability using the discrimination efficiency curve: (**a**) model-building CLIONE group; and (**b**) validation-building Kawasaki group. The groups were sorted in the order of incidence rate of fibrosis stages 3–4 and validated using the correlation between cumulative cases and the cumulative incidence of fibrosis stages 3–4. Specifically, the X-axis represents the ratio of patients in the order of groups predicting fibrosis stages 3–4, and the Y-axis represents the cumulative patients with fibrosis stages 3–4. The discrimination efficiency and stability of the curve of the model-building group were high.

**Table 1 diagnostics-15-01591-t001:** Characteristics of clinical parameters of model-building (CLIONE study) and validation groups (Kawasaki study) with ALT levels ≤ 30 U/L.

	All Patients	CLIONE Study	Kawasaki Study
	N = 289	N = 115	N = 174
Age, years	62 (18–82)	63 (28–78)	61 (18–82)
Male sex %	36	29	39
Stage, 0/1/2/3/4	78/97/51/48/15	40/39/17/13/6	38/58/34/35/9
Lobular inflammation, 0/1/2/3	33/202/48/6	9/85/18/3	24/117/30/3
Steatosis, 0/1/2/3	26/205/50/8	2/97/11/5	24/108/39/3
Ballooning, 0/1/2	147/119/23	56/46/13	91/73/10
Type-2 DM (yes/no, %)	151/138 (52)	56/59 (49)	95/79 (54)
Hypertension (yes/no, %)	130/159 (45)	65/50 (56)	65/109 (37)
Dyslipidemia (yes/no, %)	206/83 (71)	65/50 (56)	141/33 (81)
ALT (U/L)	22 (9–30)	24 (14–30)	21 (9–29)
AST (U/L)	23 (12–77)	24 (14–77)	23 (12–60)
γ-GTP (U/L)	32 (9–440)	35 (9–440)	30 (11–294)
Total cholesterol (mg/dL)	191 (90–307)	194 (117–273)	190 (90–307)
Platelet counts (10^4^/µL)	20.5 (3.4–46)	21.1 (3.4–46)	20.3 (6.6–37)
Albumin (g/dL)	4.2 (2.5–5.4)	4.2 (2.6–5.1)	4.2 (2.5–5.4)
HbA1c (%)	6 (3.5–12)	6 (3.5–9.2)	6 (3.9–12)
FBS (mg/dL)	102 (66–300)	103 (81–249)	101 (66–300)
FIB-4 Index	1.4 (0.3–12.4)	1.42 (0.38–12.4)	1.48 (0.30–7.3)

γ-GTP, γ-glutamyl transpeptidase; ALT, alanine aminotransferase; AST, aspartate aminotransferase; DM, diabetes mellitus; FIB-4, fibrosis-4; HbA1c, glycated hemoglobin; FBS, fasting blood sugar.

**Table 2 diagnostics-15-01591-t002:** Comparison of clinical parameters of patients with MASLD and stages 0–2 vs. 3–4.

	Stages 0–2	Stages 3–4	*p*-Value
	N = 226	N = 63	
Age, years	60 (18–77)	67 (33–82)	<0.0001
Male sex (%)	37	26	0.1128
Stage, 0/1/2/3/4	78/97/51/0/0	0/0/0/48/15	<0.0001
Lobular inflammation, 0/1/2/3	33/171/19/3	0/31/29/3	<0.0001
Steatosis, 0/1/2/3	18/164/39/5	8/41/11/3	0.468
Ballooning, 0/1/2	132/82/12	15/37/11	<0.0001
Type-2 DM (yes/no, %)	104/122(46)	47/16 (74)	<0.0001
HT (yes/no, %)	89/137(39)	41/22 (65)	0.0003
Dyslipidemia (yes/no, %)	162/64(71)	44/19 (69)	0.776
ALT (U/L)	22 (10–30)	21 (9–29)	0.3547
AST (U/L)	23 (12–77)	27 (15–69)	<0.0001
γ-GTP (U/L)	31 (9–440)	36 (13–228)	0.2024
Total cholesterol (mg/dL)	197 (90–307)	180 (102–279)	0.0008
Platelet counts (10^4^/µL)	21 (7.5–46)	16.2 (3.4–40.3)	<0.0001
Albumin (g/dL)	4 (2.5–5.4)	4 (2.8–4.8)	<0.0001
HbA1c (%)	5.9 (3.5–12)	6.2 (4.3–8.6)	<0.0001
FBS (mg/dL)	100 (66–300)	110 (70–195)	0.0444
FIB-4 Index	1.3 (0.3–7.3)	2.58 (0.85–12.4)	<0.0001

Overall, 289 patients with MASLD with ALT levels ≤ 30 U/L in both cohorts were older than those with stages 0–2, with significant differences in age, DM, hypertension, AST levels, platelet counts, albumin levels, HbA1c levels, and the FIB-4 index. γ-GTP, γ-glutamyl transpeptidase; ALT, alanine aminotransferase; AST, aspartate aminotransferase; DM, diabetes mellitus; FIB-4, fibrosis-4; HbA1c, glycated hemoglobin A1c; FBS, fasting blood sugar.

**Table 3 diagnostics-15-01591-t003:** Multivariable logistic regression analysis of factors associated with stages 3–4.

	Odds Ratio	95% CI	*p*-Value
Age ≥ 65 years	2.3	1.29–4.27	0.0052
Type-2 DM (yes)	3	1.55–5.80	0.0013
Hypertension (yes)	2	1.09–3.78	0.0263

Multivariate analysis revealed that the presence of DM and hypertension were significant factors. CI, confidence interval; DM, diabetes mellitus.

## Data Availability

The original contributions presented in the study are included in the article, further inquiries can be directed to the corresponding author.

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
