# Peer review of "Tips for Hepatologist Referral of Patients with Metabolic Dysfunction-Associated Steatotic Liver Disease with Alanine Aminotransferase Levels ≤ 30 U/L"

_diagnostics, 2025, doi:10.3390/diagnostics15131591_

Round 1
Reviewer 1 Report
Comments and Suggestions for Authors
This is a very well designed, performed and presented study, presenting the algorithm for diagnostics of patients with MASLD and ALT within the normal range.
The methodology is well decribed and presented. The study group is quiet large.
The results are clearly presented.
The results are of great clinical importance.
Author Response
We sincerely thank Reviewer 1 for the positive and encouraging comments on our manuscript.
We truly appreciate your recognition of the study’s design, the clarity of the methodology and results, and the clinical relevance of our findings.
Your supportive evaluation reinforces the importance of our work and motivates us to continue investigating diagnostic strategies for MASLD, particularly in patients with normal ALT levels.
Thank you once again for your thoughtful and generous review.
Reviewer 2 Report
Comments and Suggestions for Authors
This is an interesting and clinically relevant study that investigates the prevalence of advanced fibrosis in patients with MASLD (Metabolic dysfunction-associated steatotic liver disease) who present with normal transaminase levels. The study highlights an important clinical message: a significant proportion of patients with MASLD and normal ALT levels may still harbor advanced fibrosis, particularly those with older age, type 2 diabetes, and hypertension.
Strengths
The focus on a frequently overlooked subgroup—MASLD patients with normal ALT—is timely and important for improving clinical risk stratification.
The sample size is reasonable, and the inclusion of two independent cohorts strengthens the findings.
The study presents a valuable call for the reconsideration of ALT-based thresholds in clinical decision-making.
Major Concerns
Retrospective Design and Inclusion Period too long
The study is retrospective and spans a long inclusion period, which may introduce variability in biopsy indications, clinical management, and diagnostic tools. This should be acknowledged as a limitation and addressed in the discussion.
Biopsy Indications Not Explained
It remains unclear why patients with normal ALT levels underwent liver biopsy. The authors should clearly describe the rationale—e.g., imaging findings suggestive of fibrosis, clinical suspicion due to metabolic comorbidities, or protocol-driven assessment. This is essential to reduce concerns about selection bias.
Ethical Approval
The manuscript does not indicate whether the study received ethics committee approval, especially regarding liver biopsies in patients without elevated liver enzymes. This information is crucial and should be added to the methods section.
Minor Concerns
Duplicate Citations
References 5, 6, and 7 are essentially the same Delphi consensus statement published in different journals. Only one of these should be retained to avoid redundancy.
Recommendation
Major revision.
The study is of high clinical interest and contributes meaningfully to the understanding of fibrosis risk in MASLD. However, the manuscript requires clarification of methodological aspects, particularly regarding biopsy indications, ethical approval, and referencing, before it can be considered for publication.
Author Response
Authors2
We sincerely thank the reviewer for the thorough and constructive comments, which have been extremely helpful in improving the quality and clarity of our manuscript. We appreciate the recognition of the clinical relevance of our study and have carefully addressed all the major and minor concerns raised. Below, we provide point-by-point responses and detail the revisions made accordingly.
Major Concerns
1.Retrospective Design and Inclusion Period too long
The study is retrospective and spans a long inclusion period, which may introduce variability in biopsy indications, clinical management, and diagnostic tools. This should be acknowledged as a limitation and addressed in the discussion.
Response1:This study has several limitations. First, the retrospective design and the relatively long inclusion period may raise concerns about variability in clinical decision-making and diagnostic practices. However, to minimize such heterogeneity, all liver biopsy specimens were evaluated by two experienced liver pathologists using consistent histological criteria throughout the study period. This centralized and standardized pathological assessment helps ensure uniformity in fibrosis staging and inflammatory grading, thereby strengthening the reliability of our findings. Nonetheless, some degree of variation in biopsy indications and clinical management may remain due to the real-world nature of the study.
We added P8, L220-223 [Third, as a retrospective study with a long inclusion period, there may have been variability in clinical decisions and biopsy indications. However, all liver specimens were evaluated by two experienced hepatopathologists using consistent histological criteria, which reduced diagnostic variability and enhanced the reliability of histological assessments.]
2.Biopsy Indications Not Explained
It remains unclear why patients with normal ALT levels underwent liver biopsy. The authors should clearly describe the rationale—e.g., imaging findings suggestive of fibrosis, clinical suspicion due to metabolic comorbidities, or protocol-driven assessment. This is essential to reduce concerns about selection bias.
Response: Thank you for your insightful comment regarding the rationale for performing liver biopsy in patients with normal ALT levels. As the reviewer rightly pointed out, this is a crucial issue in addressing potential selection bias. In response, we have clarified this point in the revised Discussion section, where we note that possible reasons for performing liver biopsy despite ALT ≤30 U/L include mild fibrosis, mild inflammation, or suspected progressive fibrosis based on clinical assessment. Although imaging findings and medication history were not available in the current database, in clinical practice, liver biopsy was often performed when there was clinical suspicion of advanced disease due to metabolic risk factors or when fibrosis was suggested by non-invasive assessments. We have now emphasized this rationale to provide greater context for biopsy indications and to address concerns regarding selection bias.
We added P8, L224-228 [In this study, some patients underwent liver biopsy despite having ALT levels ≤30 U/L. Although imaging data and medication history were not available in the database, biopsy was typically considered when there was clinical suspicion of advanced fibrosis or steatohepatitis based on metabolic risk factors, laboratory data, or non-invasive assessments. These factors likely contributed to the clinical decision to proceed with histological evaluation, even in the presence of normal ALT levels.
3.Ethical Approval
The manuscript does not indicate whether the study received ethics committee approval, especially regarding liver biopsies in patients without elevated liver enzymes. This information is crucial and should be added to the methods section.
Response3: Thank you for pointing out the lack of information regarding ethical approval. We apologize for this omission.We added P4, L129-131 [The study protocol complied with the guidelines outlined in the 1975 Helsinki86 Declaration (as revised in Fortaleza, Brazil, in October 2013). The study was approved by the Institutional Review Board of Saga University and Kawasaki Medical School(approval no. 2020-04-R-02 and 3864).]
Minor Concerns
Duplicate Citations
References 5, 6, and 7 are essentially the same Delphi consensus statement published in different journals. Only one of these should be retained to avoid redundancy.
Response4: Thank you for pointing this out. We agree that References 5, 6, and 7 referred to the same Delphi consensus statement published in different journals. In the revised manuscript, we have retained only one representative citation and removed the redundant references to avoid duplication.
Round 2
Reviewer 2 Report
Comments and Suggestions for Authors
The aim of this study was to identify predictors of advanced liver fibrosis (stage ≥3) in patients with MASLD and ALT levels ≤30 U/L. To achieve this, two cohorts of patients who had undergone liver biopsy were analyzed: the CLIONE cohort (n=115) and the validation cohort from Kawasaki Medical Center (n=174), both collected over an extended period from 1994 to 2021. While such a broad inclusion period could raise concerns about diagnostic reliability due to potential clinical heterogeneity, the need to gather a sufficient number of patients with specific characteristics (MASLD and normal ALT) justifies this approach. Furthermore, histological review by two expert liver pathologists helped to minimize potential bias.
In this study, liver biopsy was justified based on clinical suspicion of advanced fibrosis. However, the specific reasons for performing the biopsy are not entirely clear, as the data were collected from multiple centers.
Although ALT elevation has traditionally been considered a referral criterion for hepatology evaluation, the authors show that a significant proportion of patients with ALT ≤30 U/L (up to 22%) had advanced fibrosis. This finding challenges the exclusive use of ALT as a severity marker and underscores the need to incorporate other metabolic risk factors when selecting patients for specialist referral.
The study identifies two key predictors of advanced fibrosis in this group: a FIB-4 index ≥2.67 and the presence of type 2 diabetes mellitus. The combination of these factors identified 70% of cases with fibrosis stage ≥3, while no advanced fibrosis was found among patients without diabetes and with FIB-4 <2.67. This observation has clear clinical utility, offering a simple and non-invasive method to identify MASLD patients with normal ALT who should be referred to hepatology for further assessment, including possible biopsy.
Moreover, the study is strengthened by a rigorous external validation showing a high correlation between cohorts (R² = 0.90), supporting the robustness and clinical applicability of the proposed model. The use of decision-tree analysis facilitates practical interpretation of the findings and makes it easy to integrate into referral algorithms in primary care settings.
In conclusion, this work not only emphasizes the importance of not underestimating the risk of advanced fibrosis in MASLD patients with normal ALT levels but also provides concrete tools—FIB-4 and diabetes status—to optimize specialist referral